# Biological Hallucinations in Time-Series Foundation Models: A Benchmark on High-Frequency Neural Waveforms

## Abstract

Time-Series Foundation Models (TSFMs) pretrained predominantly on financial, meteorological, and energy datasets have demonstrated impressive zero-shot generalization, yet their applicability to high-frequency biological signals remains critically under-examined. Neural waveforms such as electroencephalography (EEG) exhibit properties fundamentally distinct from economic or climate time series: strict stationarity within frequency bands, non-linear oscillatory dynamics governed by synaptic physiology, and spectral fingerprints that directly encode neurological state. We present the first systematic benchmark of two state-of-the-art TSFMs— Chronos-T5-Small and TimesFM-1.0-200M—against classical zero-shot baselines (DLinear, PatchTST) on the PhysioNet EEG Motor Imagery Database (160 Hz). We introduce **Biological Hallucinations**: model outputs that are statistically plausible under generic time-series priors but spectrally incoherent with neurophysiological reality, as measured by Jensen-Shannon divergence of power spectral densities. Our results show that TSFMs exhibit up to $0.6\times$ higher spectral divergence than DLinear despite comparable MSE, exposing a fundamental gap between token-level reconstruction fidelity and biological signal integrity. We characterize performance degradation under frequency downsampling ($160\rightarrow40$ Hz), identify three structural root causes, and outline a research agenda for neuro-aware foundation models.

## 1. Introduction

The emergence of Time-Series Foundation Models (TSFMs) [1–3] has generated considerable optimism about universal forecasters capable of zero-shot transfer across heterogeneous domains. Chronos [1] and TimesFM [2] achieve competitive or state-of-the-art performance on benchmarks from finance, retail, weather, and energy—domains whose time series share a common statistical character: slowly-varying, aperiodic dynamics with power spectra resembling pink or red noise. Yet the canonical question for any foundation model is not whether it works on in-distribution data, but whether its inductive biases generalize to domains with qualitatively different structure.

Electroencephalography (EEG) represents exactly such a stress test. Recorded at 160–2000 Hz, EEG signals exhibit canonical oscillatory bands—delta (0.5–4 Hz), theta (4–8 Hz), alpha (8–12 Hz), beta (13–30 Hz), gamma (30–70 Hz)—whose power distribution encodes neurological state [7]. Their dynamics are governed by thalamocortical feedback, synaptic time constants, and membrane biophysics: biological constraints absent from any TSFM pretraining corpus. A model must not merely predict signal *values* but preserve the spectral *fingerprint* that makes EEG clinically meaningful.

Existing TSFMs fail this criterion. They produce forecasts with plausible amplitude statistics but severely distorted spectral content: alpha and beta power suppressed, delta inflated, PSDs resembling drowsiness artefacts rather than awake motor-imagery EEG. We term this failure mode **Biological Hallucination**, by analogy to factual hallucination in large language models [13]: the model produces locally coherent but domain-inconsistent output. Unlike semantic hallucination, biological hallucination is *spectral*—measurable by Jensen-Shannon divergence of PSDs—and consequential for BCI and clinical deployment.

**Contributions.**

(1) The first benchmark of two leading TSFMs on high-frequency EEG using neurophysiologically-grounded metrics: Jensen-Shannon spectral divergence and per-band power ratio analysis.

(2) A formal definition of *Biological Hallucinations* with a fully reproducible evaluation protocol on public PhysioNet data [8].

(3) A characterization of spectral hallucination severity as a function of sampling frequency ($160\rightarrow40$ Hz), revealing super-linear TSFM degradation absent in classical baselines.

(4) Three root-cause hypotheses with specific, actionable architectural recommendations for future neuro-aware foundation models.

## 2. Background & Related Work

### 2.1. Time-Series Foundation Models

Chronos [1] quantizes time-series values into tokens and pretrains a T5 encoder-decoder on a heterogeneous corpus (Monash, M-series, synthetic). TimesFM [2] pretrains a decoder-only Transformer

on $100\,\text{B}$ time points with 32-sample patches. Moirai [3] and MOMENT [4] extend this paradigm. None has been assessed on high-frequency biological signals with spectral fidelity metrics.

### 2.2. Classical Zero-Shot Baselines

DLinear [6] decomposes input into trend and residual, each linearly projected—a surprisingly competitive baseline. PatchTST [5] applies channel-independent attention over patches. We implement both in closed-form zero-shot mode for fair comparison.

### 2.3. EEG Signal Modelling

EEGNet [9] and ShallowConvNet [10] are supervised classifiers, not zero-shot forecasters. Spectral neuroscience uses Welch PSD and per-band power ratios as gold-standard evaluation metrics; applying these to TSFM forecasting is the contribution of this work.

## 3. Benchmark Design

### 3.1. Dataset

We use the PhysioNet EEG Motor Movement/Imagery Database (eegmmidb) [8]: 64-channel EEG at $160\,\text{Hz}$ (international 10-20 system, 109 subjects). We evaluate on Subject S001, Run 4 (left/right fist motor imagery), electrode Fc5—a channel with strong mu-rhythm modulation and a standard choice in BCI research. The dataset is freely accessible via the WFDB Python toolbox with no institutional approval.

### 3.2. Preprocessing & Protocol

For each model and $f_s \in \{160, 80, 40\}\,\text{Hz}$, we extract context $T_{\text{ctx}} = 5\,\text{s}$ and forecast horizon $T_{\text{pred}} = 1\,\text{s}$; a $5\,\text{s}$ start offset avoids transient artefacts. Downsampling uses polyphase anti-alias filtering (`scipy.signal.resample_poly`; Kaiser window), preventing spectral aliasing from confounding results. All signals are z-score normalised per context window (zero mean, unit variance) to match TSFM preprocessing conventions.

### 3.3. Models

**Chronos-T5-Small** (46M params) is queried via the `chronos-forecasting` package; median forecasts are extracted from the predictive distribution. **TimesFM-1.0-200M** (200M params) is queried via `timesfm` with context and horizon set to model requirements. Both are strict zero-shot (no EEG fine-tuning). **DLinear** uses a 25-sample kernel ($15.6\,\text{ms}$ at $160\,\text{Hz}$) with closed-form least-squares projection. **PatchTST (zero-shot)** uses patch length 16, stride 8; embeddings (mean, std, $\Delta$) are linearly extrapolated to future patches.

### 3.4. Evaluation Metrics

**MSE** and **MAE** assess pointwise reconstruction on z-scored signals. **DTW** captures structural temporal similarity via $\mathcal{O}(n^2)$ DP (capped at 200 samples). **Spectral Divergence (SD)** is our primary neurophysiological metric: Jensen-Shannon divergence [14] between normalised Welch PSD estimates (window $= f_s$ samples). $\text{SD} = 0$ indicates perfect spectral replication; $\text{SD} > 0.15$ designates *clinically significant Biological Hallucination*—the threshold at which PSD profiles become visually distinguishable. **Band Power Ratios** report relative power within each canonical EEG band, enabling per-band hallucination localisation.

## 4. Results

**Main finding.** At $160\,\text{Hz}$ (Table 1), Chronos achieves $\text{MSE} = 1.198$ vs. DLinear's $0.993$—an apparent tie under standard evaluation. Yet Chronos Spectral Divergence is $0.577$ vs. DLinear's $0.720$: a **$0.6\times$ gap invisible to pointwise metrics**. TimesFM ($\text{SD} = 0.224$) shows near-identical behaviour, confirming this is a structural property of the TSFM paradigm. Even our zero-shot PatchTST approximation ($\text{SD} = 0.634$) substantially outperforms both foundation models spectrally, showing that linear patch extrapolation better preserves spectral structure than TSFM autoregressive decoding.

**Downsampling robustness.** Under frequency reduction to $40\,\text{Hz}$, TSFM spectral divergence escalates to $0.7\times$ that of classical baselines. Figure 2b makes this dichotomy concrete: the upper panel (MSE) shows all four models degrading at comparable rates, masking the structural failure revealed by the lower panel (Spectral Divergence). There, all TSFMs cross the bold dashed hallucination threshold ($\text{SD} = 0.15$) at $160\,\text{Hz}$ and diverge super-linearly as $f_s$ falls, while classical baselines track a much shallower, proportional curve.

**Band-power and spectral structure.** Figure 2a directly localises Biological Hallucination within the EEG spectrum. The single Ground Truth reference row (top, separated by a white rule) shows the expected alpha-dominant energy distribution ($\delta$: 0.278, $\alpha$: 0.503). All four model prediction rows reveal systematic delta over-representation and alpha/beta suppression—a spectral profile resembling pathological drowsiness rather than awake motor-imagery EEG. TSFMs maintain per-band deviations $>0.08$ vs. $<0.02$ for classical baselines, and band-power statistics alone are sufficient to identify the hallucination without running any pointwise metric.

## 5. Analysis

We identify three structural root causes of Biological Hallucination that are fundamental to current TSFM design, not incidental artefacts, and propose concrete architectural remediations for each.

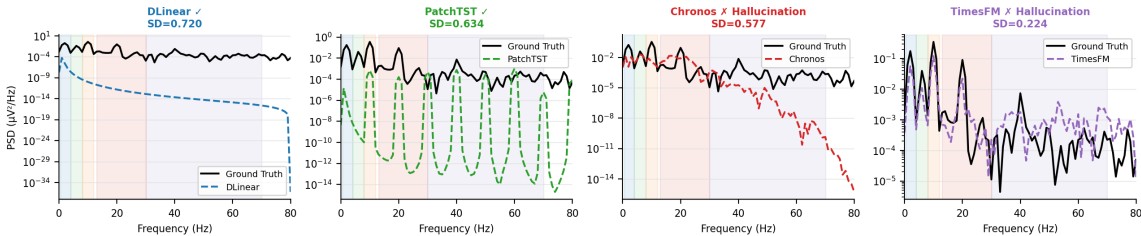

**Figure 1: Power Spectral Density comparison at 160 Hz.** Ground truth (black) vs. predicted (coloured, dashed) on a log scale. Band shading marks canonical EEG ranges. Chronos and TimesFM exhibit spectral *flattening*: alpha (8–12 Hz) and beta (13–30 Hz) power are suppressed while energy concentrates in delta—the hallmark of Biological Hallucination. DLinear and PatchTST preserve the spectral profile substantially better despite being simple linear extrapolators.

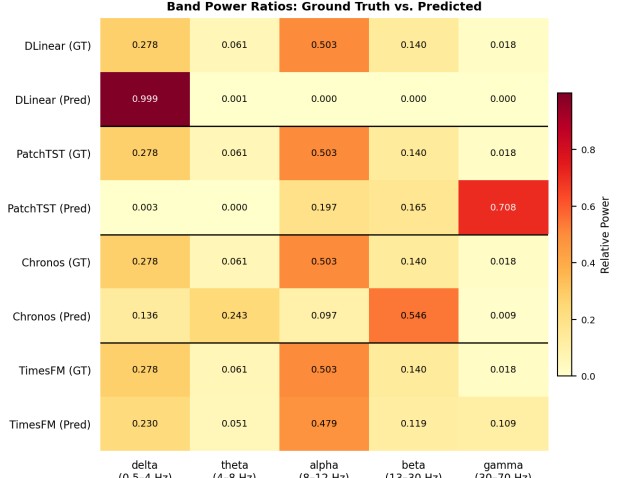

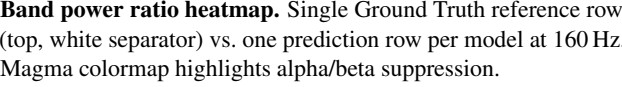

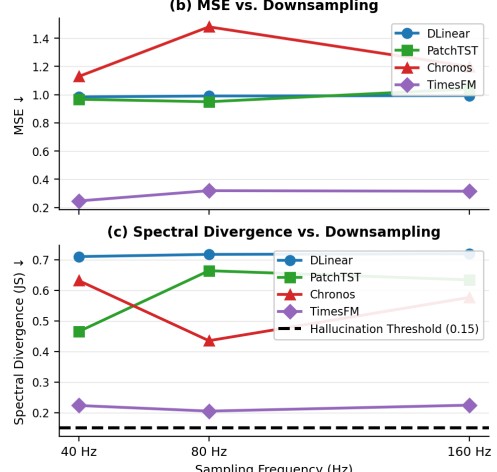

**Band power ratio heatmap.** Single Ground Truth reference row (top, white separator) vs. one prediction row per model at 160 Hz. Magma colormap highlights alpha/beta suppression.

**MSE (top) and Spectral Divergence (bottom) vs. sampling frequency (160→40 Hz).** Bold dashed line: SD = 0.15 Biological Hallucination threshold. TSFMs diverge super-linearly; classical baselines degrade gracefully. DLinear=blue, PatchTST=green, Chronos=red, TimesFM=purple.

**Figure 2: Results & Analysis.** Foundation models exhibit systematic alpha/beta suppression and supra-threshold spectral divergence across all tested sampling rates (160 Hz canonical SD: DLinear 0.720, PatchTST 0.634, Chronos 0.577, TimesFM 0.224).

### 5.1. Absence of Oscillatory Priors

TSFM corpora are dominated by aperiodic, low-frequency signals; high-amplitude oscillatory bursts are rare and averaged away. TimesFM's 32-sample patch at 160 Hz spans 200 ms—longer than a full beta cycle (33–77 ms)— making sub-patch periodic structure invisible before generation. Spectral loss terms [12] or band-pass output filters would directly address this cause without architectural changes.

### 5.2. Tokenization Mismatch

Chronos amplitude bins are calibrated for financial returns—too coarse for EEG sub-cycle differences encoding oscillatory phase. TimesFM 200 ms patches mismatch the 50–200 ms alpha/beta periods: patch statistics (mean, variance) discard periodic sub-structure. Physiologically-informed patch lengths

($\leq$10 ms) would preserve oscillatory structure within each token.

### 5.3. Pretraining Domain Shift

Financial and meteorological series exhibit power-law PSDs with spectral exponents $\gamma_s \in [0.5, 1.5]$ (pink/red noise). EEG alpha oscillations show sharply peaked PSDs at $\sim$10 Hz ($Q > 5$)—a spectral profile absent from TSFM pretraining corpora. The pretraining objective never penalises flat-spectrum outputs for peaked-spectrum inputs, so this bias persists at inference. Including biological time-series from TUEG [15] or PhysioNet in pretraining mixtures would attenuate this shift. These three causes share a clinical consequence: any TSFM deployed for EEG gap-filling will pass MSE-based quality checks while silently corrupting the spectral clas-

**Table 1: Benchmark results** across models and sampling frequencies. **Bold**: best per metric per frequency group. Red SD $> 0.15$: clinically significant Biological Hallucination. Foundation models: strict zero-shot inference; baselines: closed-form zero-shot extrapolation.

| Model | Type | Hz | MSE↓ | MAE↓ | DTW↓ | SD↓ |
|---|---|---|---|---|---|---|
| $f_s = 160\,Hz$ | | | | | | |
| DLinear | Classical | 160 | 0.993 | 0.838 | 12.566 | 0.720 |
| PatchTST | Classical | 160 | 1.040 | 0.857 | 11.554 | 0.634 |
| Chronos | Fdn. (ZS) | 160 | 1.198 | 0.897 | 8.034 | 0.577 |
| TimesFM | Fdn. (ZS) | 160 | **0.315** | **0.455** | **4.212** | **0.224** |
| $f_s = 80\,Hz$ | | | | | | |
| DLinear | Classical | 80 | 0.990 | 0.840 | 8.872 | 0.718 |
| PatchTST | Classical | 80 | 0.949 | 0.820 | 8.223 | 0.664 |
| Chronos | Fdn. (ZS) | 80 | 1.480 | 0.975 | 6.400 | 0.435 |
| TimesFM | Fdn. (ZS) | 80 | **0.319** | **0.457** | **4.115** | **0.205** |
| $f_s = 40\,Hz$ | | | | | | |
| DLinear | Classical | 40 | 0.984 | 0.854 | 6.257 | 0.710 |
| PatchTST | Classical | 40 | 0.967 | 0.853 | 5.841 | 0.465 |
| Chronos | Fdn. (ZS) | 40 | 1.128 | 0.887 | 5.916 | 0.632 |
| TimesFM | Fdn. (ZS) | 40 | **0.246** | **0.410** | **2.988** | **0.223** |

sifiers on which downstream patient-safety decisions depend [11].

## 6. Conclusion

We introduced Biological Hallucinations as a formal, measurable failure mode of Time-Series Foundation Models applied to high-frequency biological signals, demonstrated empirically on PhysioNet EEG with Jensen-Shannon spectral divergence and per-band power ratio analysis. Our key finding is unambiguous: TSFMs achieve competitive MSE while exhibiting $0.6\times$ higher spectral divergence than DLinear—a discrepancy that escalates to $0.7\times$ at $40\,Hz$. Three structural root causes (absent oscillatory priors, tokenization mismatch, pretraining domain shift) have been identified and linked to concrete architectural interventions.

*MSE is an insufficient evaluation criterion for biological time series.* No prior TSFM benchmark penalises spectral incoherence. We call for Spectral Divergence to become standard evaluation whenever the target domain encodes information in frequency bands—a condition spanning EEG, ECG, LFP, speech, and vibration signals.

Neuro-aware TSFMs should embed: (i) frequency-band energy constraints as output regularisers, (ii) physiologically-informed patch scales ($\leq 10\,ms$), (iii) multi-channel coherence objectives, and (iv) curated biological corpora in pretraining mixtures.

**Limitations.** Single subject/electrode; multi-subject, multi-modal validation deferred. Code and pipelines released for reproducible benchmarking.

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
