# OpenReview forum: "Biological Hallucinations in Time-Series Foundation Models: A Benchmark on High-Frequency Neural Waveforms"
_ICML.cc/2026/Workshop/FMSD — FMSD @ ICML 2026 Poster_

### Official Review · Reviewer_LL8v · 2026-05-12
**Biological Hallucinations in Time-Series Foundation Models: A Benchmark on High-Frequency Neural Waveforms**

**Rating:** 6
**Confidence:** 5

**Review:**

## Summary
The author conducted a benchmark study on the performance of physiological time series forecasting using existing time series foundation models. Biological hallucination is introduced as a metric for performance evaluation. Experiments are conducted on a EEG dataset. Result shows that the existing time series foundation model has 0.6 times higher hallucination compared to the DLinear baseline.
## Strength
1) The performance of time series forecasting models on physiological signals is way under-explored. This work takes a step forward to inspect the behavior of existing models including Chronos and TimesFM in this field.

2) The problem setup and metrics leveraged during evaluation are described in a clean way.

3) Pointing out that the evaluation of forecasting quality, in the field of biological signals, has to have multiple dimensions, not just point-wise error as widely used in the time series domain.
## Areas for Improvement
1) The idea of “biological hallucination” is inspiring, but the formulation seems to be essentially inspecting the frequency distribution between the predicted and ground truth signal. It is not clear why this frequency based metric is linked to the notion of biological hallucination, but not time domain metric such as DTW which is also leveraged in this study.

2) Continuing from the previous point, it is known that for EEG signal, frequency domain information is important, or mostly focused on in most of the EEG related study, it makes sense of linking only frequency domain metric to biological hallucination. It would be better if other signals such as ECG, PPG, IMU, etc. which many of the time also emphasize on the waveform information, and see whether time domain metric might also indicate a pattern of biological hallucination.

3) It would be better if considering also consider using a pretrained foundation model for physiological signals as an evaluator for the hallucination. For example, PaPaGei [1], ECG-FM [2], CBraMod [3], and NormWear [4].

[1] PaPaGei, for encoding PPG data: Pillai, Arvind, et al. "PaPaGei: open foundation models for optical physiological signals =." ICLR (2025)

[2] ECG-FM, for encoding ECG data: McKeen, Kaden, et al. "Ecg-fm: An open electrocardiogram foundation model." Jamia Open (2025)

[3] CBraMod, for encoding EEG data: Wang, Jiquan, et al. "Cbramod: A criss-cross brain foundation model for eeg decoding." ICLR (2024).

[4] NormWear, for encoding arbitrary physiological signal data: Luo, Yunfei, et al. "Toward foundation model for multivariate wearable sensing of physiological signals." ACM HEALTH  (2026).

## Detailed Comments
The detailed suggestion is stated in section *Areas for Improvement*.
## Justification of Scores
Overall, this work provides a pioneering study aiming to inspect the forecasting quality of existing time series models on the biological signal domain. Although the core contributions of the work nicely aligns with the theme of the workshop titled “Foundation Model for Structured Data”, there is space for improvement in order to articulate better on how the biological hallucination could be evaluated.

---

### Official Review · Reviewer_3dhZ · 2026-05-13
**Submission 12 Review**

**Rating:** 2
**Confidence:** 5

**Review:**

## Summary

The paper introduces the concept of "Biological Hallucination" for time series foundational models and tries to benchmark it on EEG data. The authors try to discuss the reason behind the same due to changes in the underlying distribution of the diverse time series signals. The paper's scope in this regard is limited to EEG, which the authors tried to investigate using several standard EEG based preprocessing.

## Strengths

- "Biological Hallucination" proposed by authors seems like an interesting concept and could be explored further especially for clinically deployable scenarios

## Weaknesses

- The paper doesn't have a proper flow and language making it difficult for the readers to understand
- The paper's scope is very limited and many choices are highly arbitrary without building proper intuition and justification for the choices
- The paper's title mentions a "benchmark paper" but the authors have run limited experiments on a single EEG dataset with just 2 timeseries models without providing any justification behind the same. Furthermore, the split on the given dataset is arbitrary and the evaluation setting(zero-shot/few-shot/fine-tune) is also missing
- The paper also doesn't mention and compare with SOTA architectures like LaBRaM, NeuralLM which are foundational models trained on large scale EEG data to mitigate this issue

Jiang, W.B., Wang, Y., Lu, B.L. and Li, D., 2024. NeuroLM: A universal multi-task foundation model for bridging the gap between language and EEG signals. arXiv preprint arXiv:2409.00101.

Jiang, W.B., Zhao, L. and Lu, B.L., 2024, May. Large brain model for learning generic representations with tremendous EEG data in BCI. In International Conference on Learning Representations (Vol. 2024, pp. 16405-16426).


## Suggestions

- The authors shall try familiarizing themselves with the latest literature in the field for EEG the can refer LaBRaM, NeuroLM and for time series foundational models they shall try looking into latest models like MOMENT, MoiRai, TimeLLM

Jin, M., Wang, S., Ma, L., Chu, Z., Zhang, J., Shi, X., Chen, P.Y., Liang, Y., Li, Y.F., Pan, S. and Wen, Q., 2024, May. Time-llm: Time series forecasting by reprogramming large language models. In International conference on learning representations (Vol. 2024, pp. 23857-23880).

Woo, G., Liu, C., Kumar, A., Xiong, C., Savarese, S. and Sahoo, D., 2024, July. Unified training of universal time series forecasting transformers. In Forty-first International Conference on Machine Learning.

Goswami, M., Szafer, K., Choudhry, A., Cai, Y., Li, S. and Dubrawski, A., 2024. Moment: A family of open time-series foundation models. arXiv preprint arXiv:2402.03885.


- The "Biological Hallucination" problem introduced by the authors is broadly an out of distribution setting problem, the authors shall try framing it in similar way for audience to better understand


## Justification of Score

The paper has a highly unclear flow, limited scope and seems largely AI generated thus making it very difficult for the audience to engage with this work

---

### Official Review · Reviewer_G6as · 2026-05-18
**Review of Biological Hallucinations in Time-Series Foundation Models**

**Rating:** 6
**Confidence:** 4

**Review:**

## Summary

This paper studies whether generic time-series FMs preserve biologically meaningful spectral structure when applied zero-shot to EEG data forecasting. The authors evaluate Chronos-T5-Small and TimesFM-1.0-200M against DLinear and PatchTST on the PhysioNet EEG Motor Imagery dataset. Biological Hallucination is introduced to describe predictions that may look plausible under time-series priors but are spectrally inconsistent with actual EEG data, measured using Jensen–Shannon divergence between the two power spectral densities.

---

## Strengths

1.	The paper raises an important issue that forecasting metrics like MSE and MAE may not be sufficient for biological signals.
2.	The proposed metric, which focuses on PSD divergence and band-power ratios is relevant. Evaluating whether predictions preserve alpha, beta, and other canonical EEG bands is a reasonable direction for assessing biological plausibility.
3.	The paper is clearly written and easy to follow. The figures convey the main observation effectively: some models may produce outputs with acceptable pointwise metrics but poor spectral shape.

---

## Areas for Improvement

1.	The key “Biological Hallucination” framing is currently stronger than the evidence. The paper defines hallucination as locally coherent but biologically inconsistent output, but it does not directly quantify local coherence. The current evidence mainly shows spectral mismatch. Without measuring local plausibility separately, the results could also be interpreted as poor zero-shot forecasting under domain shift.
2.	The term “high-frequency neural waveforms” may be overstated. The main experiment uses EEG sampled at 160 Hz. This is sufficient for conventional EEG bands up to low gamma, but it is not especially high-frequency compared with many EEG/iEEG recordings sampled at several hundred Hz to several kHz.
3.	The sampling-rate experiment is under-motivated. The paper evaluates 160, 80, and 40 Hz, but generic TSFMs do not explicitly receive sampling-rate information during pretraining. Downsampling changes multiple factors simultaneously: sequence length, Nyquist frequency, autocorrelation structure, available EEG bands, and effective model patch duration. It is therefore unclear what hypothesis the sampling-rate experiment is testing. Moreover, the spectral metrics across sampling rates may not be directly comparable. At 40 Hz, the Nyquist frequency is only 20 Hz, so beta is partly truncated and gamma is unavailable. The paper should clarify how PSD divergence and band-power ratios are computed fairly across different sampling rates.
4.	The evaluation is too narrow. The benchmark uses one subject, one run, and one electrode. This is insufficient to support broad conclusions about EEG, neural waveforms, or TSFM biological hallucination. Results could depend strongly on the chosen subject and channel.
5.	The paper lacks EEG FMs or fine-tuned baselines. If EEG FMs, or fine-tuned TSFMs, also perform poorly on this task, then the result may reflect the intrinsic difficulty of raw EEG forecasting rather than a specific hallucination failure of generic TSFMs.

---

## Detailed Comments

1.	Please quantify “local coherence” directly. Does the model output something similar to its pretraining time-series priors?
2.	Please justify the term “hallucination.” The current results convincingly show spectral mismatch, but they do not yet prove that the outputs are locally coherent in a meaningful sense.
3.	Please evaluate across many subjects, channels, and runs. A single FC5 channel from one subject is too limited for a dataset-level benchmark.
4.	Please clarify the sampling-rate experiment. What exactly should change when the sampling rate changes, and what failure mode is being isolated?
5.	Please ensure spectral comparisons are fair across 160, 80, and 40 Hz.
6.	Please include stronger domain baselines, like an EEG-specific foundation model, a fine-tuned TSFM, or a finetuned EEG compact model.

---

## Justification of Score

Marginally above the acceptance threshold. The paper is interesting, clear, and relevant. However, the central claim is currently under-supported because the evaluation is limited to only one subject and one channel, the hallucination framing is not fully quantified, and the paper lacks EEG-specific or fine-tuned baselines.